# Effect of *Brosimum alicastrum* Foliage on Intake, Kinetics of Fermentation and Passage and Microbial N Supply in Sheep Fed *Megathyrsus maximus* Hay

**DOI:** 10.3390/ani14081144

**Published:** 2024-04-09

**Authors:** Vicente Valdivia-Salgado, Ever del Jesús Flores-Santiago, Luis Ramírez-Avilés, José Candelario Segura-Correa, Jesús Miguel Calzada-Marín, Juan Carlos Ku-Vera

**Affiliations:** 1Livestock Consultant, Estelí C.P. 31000, Nicaragua; viznat1414@yahoo.com.mx; 2Southeast University Regional Unit, Autonomous University Chapingo, Carr. Teapa-Vicente Guerrero km 7.5, Teapa C.P. 86807, Tabasco, Mexico; 3Faculty of Veterinary Medicine and Animal Science, University of Yucatan, Carr. Merida-Xmatkuil km 15.5, Merida C.P. 97100, Yucatan, Mexico; luis.ramirez@correo.uady.mx (L.R.-A.); jose.segura@correo.uady.mx (J.C.S.-C.); calzadamarinmiguel@hotmail.com (J.M.C.-M.); kvera@correo.uady.mx (J.C.K.-V.)

**Keywords:** *Brosimum alicastrum*, hair sheep, rumen fermentation, voluntary intake

## Abstract

**Simple Summary:**

The foliage of the tree *Brosimum alicastrum* is widely employed by small farmers in Latin America to supplement cattle, sheep and goats during the dry season. Foliage is manually cut, and the branches are chopped and fed directly, being highly palatable to ruminant species. The foliage remains green throughout the year, and the tree can be defoliated twice per year. Cattle can also directly browse the foliage from the tree, but it is mostly used in cut and carry systems and transported in carts to farms. The experiment hereby described showed that the crude protein fraction of the foliage of *B. alicastrum* is highly fermentable in the rumen, supplying nitrogen for growth of the microbial population. As the percent of foliage of *B. alicastrum* in the ration was increased, dry matter intake by sheep was concomitantly increased, as was the rate of the passage of digesta through the rumen and the supply of microbial nitrogen to the small intestine. The foliage of *B. alicastrum* is a valuable feedstuff for ruminants in the tropics and can be used as a cheap source of protein during the critical periods of the year to improve weight gain and milk yield in a sustainable way.

**Abstract:**

An experiment was carried out to assess the effect of the incorporation of sun-dried foliage of *Brosimum alicastrum* into rations based on hay of *Megathyrsus maximus* on intake, rumen fermentation, kinetics of passage, microbial nitrogen supply to the small intestine, apparent digestibility in Pelibuey hair sheep. Four rations were randomly allotted to four rumen-cannulated lambs (BW = 37.4 ± 4.9 kg) using a 4 × 4 Latin square design to assess the effect of increasing levels (0, 15, 30 and 45% DM basis) of foliage of *Brosimum alicastrum* on a basal ration of *M. maximus.* Organic matter intake and water consumption increased linearly (*p* < 0.01) with increasing levels of *B. alicastrum* in the ration. The rate and potential extent of rumen fermentation of OM and CP of *B. alicastrum* were 10.6%/h and 86.6% and 11.4%/h and 95.2%, respectively, but no effect (*p* > 0.05) was found on the potential rumen degradation of DM (40.2%) or on the rate of degradation of DM (0.033%/h) of *M. maximus*, although a positive effect was found in the rumen degradation rate of NDF (*p* < 0.05). VFA and ammonia concentration in the rumen and the rate of passage of solids and liquids through the rumen (*k*_1_) increased linearly (*p* < 0.01) with increasing levels of *B. alicastrum.* Rumen pH was not affected by the incorporation of *B. alicastrum* (*p* > 0.05). Microbial nitrogen supply to the small intestine (*p* < 0.001), apparent digestibility of dry matter (*p* < 0.01) and NDF (*p* < 0.05) of the rations were also significantly increased as a result of the incorporation of *B. alicastrum* foliage. Results from this experiment suggest that the foliage of *Brosimum alicastrum* can be readily incorporated at around 30% of the ration of dry matter in hair sheep with beneficial effects on feed intake, rate of passage and microbial N supply to the lower tract.

## 1. Introduction

*Brosimum alicastrum* Swartz (Spanish common name: ramon) is a tropical tree that has been used since ancient times by the Maya population of Central America for various purposes, including food, shade and forage [1,2,3]. The foliage of *Brosimum alicastrum* is widely utilized by small-scale dairy and cattle farmers in Latin America as a source of forage; its yield may reach 2 tons of DM/ha/year [4]. It is one of the few trees that remain green during the six-month dry season. However, relatively little is known regarding its nutritive value for ruminants. Medina-Figueras [5] was the first to describe the apparent digestibility of DM and CP of the foliage and seeds of *B. alicastrum* by cattle. Yerena et al. [6] determined the apparent DM digestibility (67%) of *B. alicastrum* foliage by cattle. Recently, Sarmiento-Franco et al. [7] proposed that the seeds of the *Brosimum alicastrum* tree may represent a good source of nutrients for different animal species given their high starch content (71.2%), and Soberanis-Poot et al. [8] advanced that *Brosimum alicastrum* may be considered a rational option for animal production systems in view of the need for adaptation to climate change.

The present work was carried out to assess the effect of incorporating increasing levels of sun-dried *Brosimum alicastrum* foliage into a poor-quality grass (*Megathyrsus maximus*) ration on voluntary intake, rumen digestion, kinetics of passage throughout the gastrointestinal tract, microbial N supply to the small intestine and apparent digestibility by Pelibuey hair sheep.

## 2. Materials and Methods

### 2.1. Animals

Animal care was conducted under the supervision and approval of the Academic Committee of the Faculty of Veterinary Medicine and Animal Science, University of Yucatan, according to the regulations established for the handling and welfare of experimental animals. Four entire male Pelibuey sheep (initial average BW = 37.4 ± 4.9 kg; 9 months old) fitted with rumen cannulas (7.5 cm diameter; Bar Diamond, Parma, ID, USA) were used. During the experimental period, sheep were housed indoors in metabolic crates (60 × 150 cm) to allow for the complete separation and recovery of feces and urine. They were treated against internal and external parasites and given an intramuscular injection of vitamins A, D and E before the start of the experiment. Two Zebu bulls weighing 427 ± 32 kg BW (3 years old) fitted with rumen cannulas (10 cm internal diameter, Bar Diamond Parma, ID, USA) were used to measure rumen degradation of *Brosimum alicastrum* foliage. Cattle were chosen to assess kinetics of fermentation of *B. alicastrum*, due to their larger capacity to hold incubation bags in the rumen compared to sheep.

### 2.2. Experimental Rations and Design 

Foliage of *Brosimum alicastrum* was harvested from mature trees and sun-dried on concrete floor for two days. *Brosimum alicastrum* and *Megathyrsus maximus* were ground twice in a grinder with a screen 0.5 cm diameter to avoid selection by sheep and mixed in the following ratios: 0:100; 15:85, 30:70 and 45:55 on a DM basis. In all rations, 9.4% cane molasses (DM basis) was included. Levels of incorporation of *B*. *alicastrum* were allocated on the premise that, under practical conditions, farmers would not incorporate foliage of *B. alicastrum* as more than a third of the ration. Rations were not corrected for equal N content to test the direct effect of *B. alicastrum* level on response variables. The chemical compositions of *B. alicastrum* foliage, *Megathyrsus maximus* hay and the rations are shown on Table 1. The experiment was designed as a 4 *×* 4 Latin square with four periods of 20 d (10 d for adaptation and 10 d for measurements). Bulls used to evaluate rumen degradation of *Brosimum alicastrum*, were fed hay of *Megathyrsus maximus*.

### 2.3. Voluntary Feed Intake and Digestibility

Experimental rations (Table 1) were offered ad libitum every morning at 08:00 and a 10% refusal, relative to the amount offered, was allowed. Water was freely available. Feed intake was measured as the difference between the amount offered and that refused the following day. In vivo apparent digestibility of the feed components was measured by total collection of feces at 08:00 h for 7 days. Daily feces collection was made during the last 7 days of each experimental period. Feces were weighed, bulked, mixed, and a 10% aliquot was taken and stored in plastic bags at −20 °C until analysis.

### 2.4. Rumen Fermentation 

Kinetics of rumen fermentation of DM, OM, NDF and CP of *B. alicastrum* and *M. maximus* were measured by means of the nylon bag technique [9]. Nylon bag dimensions were 5 × 7 cm for sheep and 10 × 20 cm for bulls, with pore size of 53 µm (Bar Diamond, Parma, ID, USA). A total of 3 grams of sample were placed into each bag and incubated in the rumen for 6, 9, 12, 24, 48, 72 and 96 h in sheep, and 5 g of sample were placed into each bag and incubated in the rumen at the same incubation times in bulls. Samples (nylon bags) were introduced by triplicate into the rumen. After incubation, bags were washed in a washing machine until the water was clear and dried in a forced-air oven at 60 °C for 72 h. Rumen disappearance data were fitted into the model *p* = *a* + *b* (1 − e^−*ct*^) [10] to estimate degradation constants *a*, *b* and *c*, where *p* is percent degradation at time *t*; *a* is the zero time intercept; *b* represents the asymptote to the equation; *c* is the fractional degradation rate of *b* and *t* is time of incubation. Effective degradability was estimated, employing the equation *a* + *bc*/(*c* + *k*), where *k* is the determined outflow rate from the rumen (per hour) [10]. The washing loss at zero time was estimated following the procedures described by López et al. [11]. 

### 2.5. Purine Derivative Excretion

Microbial N supply to the small intestine was estimated by the purine derivative excretion technique [12]. Daily urinary output was collected in about 50 mL of 1 M H_2_SO_4_ to prevent ammonia-N loss (final urine pH < 3), and every 24 h, urine collection was diluted with tap water (5:1) (to prevent precipitation of uric acid during storage), filtered through glass wool and sampled. Urine samples were stored at –20 °C before analysis of purine derivatives and total N.

The amount of microbial purines absorbed was estimated from equation below [12]. Y = 0.84X + (0.15W^0.75^ e^−0.25x^) where Y is the total (mmol/d) urinary excretion of purine derivatives; X is the exogenous absorbed purines (mmol/d); 0.84 is the proportion of purine derivatives excreted in the urine; 0.15 is the endogenous purine derivative excretion (mmol/d); W^0.75^ is the metabolic BW and 0.25 is the rate constant for the replacement of de novo synthesis of endogenous purines by exogenous purines. Daily supply of microbial N to the small intestine was estimated following the model described by Chen et al. [13], namely: 

Microbial N supply (g/d) = 70X/(0.83 × 0.116 × 1000) = 0.727X, where 70 is the N content of purines (mg/mmol), 0.83 is the digestibility of microbial purines and 0.116 is the proportion of microbial N as purine N.

### 2.6. Rumen pH, Volatile Fatty Acid and Ammonia Concentrations

Ruminal liquor was withdrawn from the rumen of each sheep using a vacuum pump at 0, 3, 6 and 9 h post-prandial for pH, VFA and ammonia determination. Rumen pH was measured in rumen liquor after withdrawal with a portable potentiometer (Cole Parmer, Vernon Hills, IL, USA) previously calibrated with buffers of pH 4 and 7. Immediately after pH determination, a sample (20 mL) of rumen liquor was filtered through four layers of cheesecloth and then treated with 20 mL of an HCl solution (0.2 M) and kept at 4 °C until analyzed for ammonia-N concentration. Concentration of ammonia (NH_3_-N) in rumen liquor was determined with a specific ion electrode (Corning Ammonia Electrode Cat. No. 476130) as suggested by Galyean and Chabot [14]. A sample (4 mL) of rumen liquor was taken after filtering, 1 mL of metaphosphoric acid solution was added, and it was kept at 4 °C until analyzed for VFA molar proportions by gas chromatography (Hewlett-Packard 5890; equipped with a flame ionization detector; the column was HP—FFAP 30 m × 0.53 mm). 

### 2.7. Rate of Passage of Liquids and Solids

Kinetics of liquid flow was estimated by administering polyethylenglycol (PEG, molecular weight 4000; Merck, Darmstadt, Germany) into the rumen. Twelve grams of PEG were dissolved in distilled water (50 mL) and infused through the rumen cannula; samples of rumen liquor were then taken at 0, 2, 4, 8, 12, 18, 24, 36 and 48 h after dosing. PEG was assayed by the turbidimetric technique [15]. Regression techniques (marker concentration vs. time) were employed to estimate the volume, outflow rate and turnover rate of liquids from the rumen of each sheep. The rate of passage of solid digesta was estimated by the Cr-mordanting technique [16]. Forty grams of Cr-mordanted fiber were introduced through the rumen cannula, and fecal samples were taken directly from the rectum at 0, 6, 8, 10, 12, 16, 24, 36, 48, 72, 96 and 120 h after dosing to describe the excretion curve of the marker. The two-compartment model [17] was used to estimate kinetics of passage through the gastrointestinal tract:Y = A e*^k^*^1(*t*−TT)^ − A e^*k*2^^(*t*−TT)^ where *k*_1_ is the rate of passage through the rumen; *k*_2_ is the rate of passage through the caecum and proximal colon and TT is the transit time through the gastrointestinal tract.

### 2.8. Chemical Analysis

Ration samples were collected daily during each of the four experimental periods and pooled across days within period for assay of DM, OM and N [18]. NDF and ADF were determined as described by Goering and Van Soest [19]. Urinary allantoin was determined by the Rimini–Schryver reaction [20] and uric acid by the technique described by Guerci [21], following the method of Caraway. Xanthine and hypoxanthine were enzymatically converted to uric acid by treating the urine with xanthine oxidase (EC 1.2.3.2) and added to the amount of uric acid [22]. Metabolizable energy concentration of the rations was estimated from the digestible organic matter in dry matter (DOMD %) based on the equation described by MAFF [23]: ME = 0.15 × DOMD %.

### 2.9. Statistical Analysis

Data were analyzed with a 4 × 4 (period x animal) Latin square design using the GLM procedure of SAS [24]. The statistical model was: $$Y_{ijk}=µ+\rho_{i}+A_{j}+{Τ}_{k}+\varepsilon_{ijk}$$
where *Y_ijk_* = Analysed variable; µ = General mean; $\rho_{i}$ = Period (*i* = 1, 2,… *k*); A*_j_* = Animal (*j* = 1, 2, … *k*); T*_k_* = Treatment (*k* = 1, 2, 3, 4); ε*_ij__k_* = Standard error.

Polynomial orthogonal contrasts were employed to analyze the linear, quadratic and cubic components of the response to the increasing levels of *Brosimum alicastrum* foliage in the rations. Data on *Brosimum alicastrum* rumen degradation were not statistically analyzed; just the means were taken.

## 3. Results

*B. alicastrum* foliage was higher in CP content, similar in DM and OM content and lower in NDF, ADF, hemicellulose and lignin content than *Megathyrsus maximus* hay (Table 1). Dry matter and OM contents of the rations were similar. The crude protein content in the rations increased, while the cell wall components decreased with graded levels of *B. alicastrum* in the rations (Table 1).

Organic matter and CP of *B. alicastrum* were rapidly and extensively degraded in the rumen (Table 2). The dry matter, OM, NDF and ME intakes increased linearly (*p* < 0.01) as the level of *B. alicastrum* in the ration was increased (Table 3). The incorporation of *B. alicastrum* did not significantly affect the potential degradation of DM or OM in the rumen of the basal ration of low-quality *Megathyrsus maximus* hay (Table 4) but increased the degradation rate of NDF (*p* < 0.05). the effective degradability of DM, OM and NDF were linearly decreased (*p* < 0.01) as the level of incorporation of *B. alicastrum* into the ration was increased. The rate of passage of solid digesta through the rumen (*k*_1_) and through the caecum and proximal colon (*k*_2_) increased linearly (*p* < 0.05), whereas the transit time (TT) through the gastrointestinal tract diminished (*p* < 0.05) as the level of incorporation of *B. alicastrum* into the ration was augmented (Table 5). The incorporation of *B. alicastrum* foliage also led to a significant (*p* < 0.01) linear increase in the turnover rate and in the outflow rate of liquids from the rumen (Table 5). No significant differences between treatments were detected for rumen volume.

The concentration of ammonia-N in the rumen was significantly (*p* < 0.05) increased by the incorporation of *B. alicastrum* foliage into the ration (Table 6). The rumen ammonia-N concentration decreased gradually postprandially across treatments (Table 6). The rumen pH was kept within normal limits for cellulolysis (6.4−6.9). There were no significant effects (*p* > 0.05) of supplementation on the VFA concentration at different times of sampling; for this reason, data were not presented. However, a significant effect was found in the VFA average concentration (*p* < 0.05), but molar proportions were not altered (*p* > 0.05).

Urinary excretion of allantoin, uric acid, xanthine and hypoxanthine (Table 7) increased linearly as the level of *B. alicastrum* in the ration was increased. Consequently, the supply of microbial N to the small intestine increased linearly (*p* < 0.001) as the level of *B. alicastrum* in the ration was increased (Table 7). Nitrogen intake, N excretion in feces and urine, as well as the N retention, increased linearly (*p* < 0.001) with the incorporation of *B. alicastrum* foliage. The apparent in vivo digestibility of DM, OM and NDF increased linearly with graded levels of *B. alicastrum* in the ration (Table 8).

## 4. Discussion

The positive effect of the incorporation of foliage of *B. alicastrum* into the ration on DM and OM intake could be explained despite the increase in the intake of degradable N in the rumen [25,26,27]. It is generally recognized that the concentration of NDF in forages is the main factor limiting dry matter intake [28,29]. In this study, the incorporation of foliage of *B. alicastrum* not only reduced the NDF content in the ration, but it also increased its rumen degradation rate.

Foliage of *Brosiumum alicastrum* increased the total DM and DM intake of the basal ration by (98%) and (52%), respectively, at the level of 30% incorporation; these values are higher than those obtained with other foliages, such as *Sesbania sesban*, *Chamaecytisus palmensis*, *Acacia agustiniana* and *Leucaena leucocephala* [30,31,32]. Another possible explanation for the high DM intake induced by *B. alicastrum* could be related to the activity and content of phenolic compounds; this forage had 2.4% of extractable phenolics, with a protein precipitating capacity (PPC) of 16 µg/mg of bovine albumin serum, values much lower compared with the 3.5% and PPC of 146 µg/mg in *Acacia pennatula* of extractable phenolics and PPC, respectively. The content of phenolic compounds and their activity may have had an influence on N utilization and OM intake [26]. 

The high rumen fermentation of OM, CP and NDF of *B. alicastrum* were similar to those found for *Sesbania sesban* by Kamatali et al. [33]. Hovell et al. [34] have shown that the high ruminal degradation of forages is generally associated with a low NDF content, as found in foliage of *Brosimum alicastrum* (37.5%). In addition, the high rate and extent of degradation of *B. alicastrum* in the rumen (Table 2) could had a positive effect on feed intake. Rumen degradation of forage has been positively associated with feed intake [25,35,36]. The particulate passage rate was increased with the inclusion of *B. alicastrum* in the ration, and this effect has been associated with intake [30,37,38]. The increase in the NDF and OM degradation rate of *Megathyrsus maximus,* as well as the high rumen degradation rate of *B. alicastrum,* could explain the positive effect on *Megathyrsus maximus* intake. Species with high ruminal degradation promote less substitutive effects [30]. On the other hand, in other species, there is little effect on intake of the basal ration, or the effect is substitutive [30,31,32]. The increase in OM intake of *Megathyrsus maximus* could have considerable practical application in the tropics, where it is important to obtain the highest efficiency of utilization of low-quality forage.

Metabolizable energy intake was 0.4, 0.93, 1.4 and 1.7 times the maintenance requirements for hair sheep compared with the ME requirement reported by Kawas and Huston [39] in hair sheep. This increase in ME intake may be associated to the higher DOM intake.

Although potential DM degradability was not affected by the incorporation of foliage of *B. alicastrum* into the ration, the degradation rate of DM, OM and NDF were increased. Trends for an increase in DM degradation rate when tree fodders are used as supplements with low-quality forages have been reported [40,41], probably associated with a high ammonia-N concentration in rumen liquor [30,40]. Bonsi et al. [31] reported the highest degradation rate, with high levels of ammonia-N in the rumen. Even when the levels of ammonia-N in all treatments were higher than those recommended by Satter and Slyter [42] and Boniface et al. [43], for maximum growth in vitro and maximum degradation in the rumen, respectively, the ration without the incorporation of *B. alicastrum* (control), with a high concentration of ammonia-N, showed a lower degradation rate. The control ration without the incorporation of *B. alicastrum* foliage was most likely deficient in rumen degradable protein, as evidenced by the low supply of microbial N to the small intestine. This result suggests that the amount of nitrogen ingested and the N:ME ratio may have influenced rumen degradation. An important role may have been played by the N:ME ratio in the degradation rate; estimations carried out following the method used by Umunna et al. [30], with N and OM slowly degradable in the rumen, gave values of 1.45 and 1.54 g of N per MJ ME, which is closer to the 1.34 g value given by ARC [44]. The reduction observed in the effective degradation of DM, OM and NDF of *Megathyrsus maximus* may be explained by the increase in the passage rate through the rumen, which leave less time available for microbial attack, since effective degradability decreases with an increase in the rate of passage from the rumen [45].

The incorporation of foliage of *B. alicastrum* into the ration augmented the ammonia-N concentration in the rumen liquor, and this may have increased the particulate passage rate [46]. On the other hand, microbial attack, chewing and rumination are responsible for the reduction in the particulate size of forage, as it is known that only particles with the critical size can pass through the rumen and into the omasum [47], as well as those particles with higher specific gravity [48]—chewing and rumination promote changes in the physical structure of forage, increasing their specific gravity [28] due to the increase in their capacity for hydration [49]. The increase observed in the ammonia-N concentration in the rumen with the incorporation of *Brosimum alicastrum* was similarly recorded in Pelibuey sheep by Alayón et al. [25], when they incorporated foliage of *Gliricidia sepium* in a basal ration of *Cynodon nlemfuensis*. A linear increase in the ammonia-N concentration was reported by Bonsi et al. [40], using *Sesbania sesban* as a supplement in sheep. In the experiment hereby described, all treatments showed values for ammonia-N higher than those suggested by Satter and Slyter [42] for maximum microbial growth under in vitro conditions and those reported by Boniface et al. [43] for maximum fermentation of OM in tropical grasses. The increase in rumen ammonia-N concentration may be due to the sgreater N intake and to the higher rumen degradation of N in *Brosimum alicastrum*.

A large volume of liquid in the rumen increases the dilution rate and rumen outflow in sheep [50]. The results of the present experiment showed that the dilution rate had a positive relationship with water consumption (r^2^ = 0.77). Furthermore, the outflow rate and turnover rate were positively related to DM intake (r^2^ = 0.75 and r^2^ = 0.83, respectively). Rations that promote high DM intake induce a high outflow rate compared to those that result in a low DM intake [31,40]. Bonsi et al. [40] found a significant effect on the kinetics of liquids in sheep supplemented with *Sesbania sesban*, which has a similar rumen degradation profile compared to foliage of *B. alicastrum*.

The incorporation of foliage of *B. alicastrum* into the ration increased the microbial nitrogen supply (MNS; g/day). Bonsi et al. [40] and Karda and Dryden [51] also reported a greater MNS (g/day) and efficiency in small ruminants supplemented with multipurpose trees. Likewise, Valdivia et al. [52] reported a higher MNS in cows in the silvopastoral system with *Leucaena leucocephala* compared with those grazing *Megathyrsus maximus* grass only. The microbial nitrogen supply to the small intestine showed a positive relation to the particulate passage rate (r^2^ = 0.87) and outflow rate (r^2^ = 0.82). Microbial protein synthesis has been related to the particulate passage rate [53], as well as the outflow rate [50,54]. Umunna et al. [55] related improvements in MNS to an increase in OM intake; in this study, MNS was related positively correlated with OM intake (r^2^ = 0.89). The increase in MNS (g/day) may be due to a better N:energy relationship in the rumen, promoted by feeding foliage of *B. alicastrum*. The rations’ soluble N:ME in the OM, estimated following the method suggested by Umunna et al. [30] were: 1.74, 1.95, 2.10 g and 2.2 g N per MJ ME for 0%, 15%, 30% and 45% levels of *B. alicastrum*. The proportions of slowly degradable N to slowly fermentable ME (g N/MJ) were: 0.68, 1.12, 1.45 and 1.54 for 0%, 15%, 30% and 45% of *B. alicastrum,* respectively. Compared with 1.34. g N per MJ ME reported by ARC [44], this suggest that ME was deficient in relation to fermentable N, in the control and 15% *B. alicastrum* rations. However, the rations slowly degradable N:OM with 0% and 15% of *B. alicastrum* had good levels of energy but not enough N; this may affect MNS. In diets with 30% and 45% of *B. alicastrum*, there was a good balance of N:ME; the small differences in the efficiency of MNS with these rations may be explained by this. Although ammonia-N concentration increased with incorporation of *B. alicastrum* in the ration and the values were higher than those suggested by Satter and Slyter [42] for maximal microbial growth, it was poorly related to MNS (r^2^ = 0.30). The values of the efficiency of MNS found in the rations supplemented with 30% and 45% *B. alicastrum* were similar to those found by Bonsi et al. [40] with *Sesbania sesban* and higher than values reported by Masama et al. [56] and Hindrichsen et al. [57] with *Leucaena leucocephala*. The differences with leucaena could be due to a lower rumen degradation and slower degradation rate of N in Leucaena. This may be influenced by phenolic compounds—low fermentation of N affects microbial nitrogen synthesis [55]. *Brosiumum alicastrum* has extractable phenolics with low PPC; for this reason, probably, *B. alicastrum* increased MNS and the efficiency even when it had a lower N content than that in the foliage of others tree species.

The better N balance in the rations incorporating foliage of *Brosimum alicastrum* may be due to the increased N intake and improvement in the efficiency of MNS. A better efficiency of MNS, along with an increase in N retention in the current study, are in agreement with the findings of Bonsi et al. [31], Masama et al. [56] and Karda and Dryden [51], which were drawn from the use of tree foliage as supplements in tropical rations. The nitrogen intake increased by 3% in the ration with 45% *Brosimum alicastrum* compared with the ration with 30%. In the same proportion, there was an increase in N retention, probably due to the good balance of N:ME, which improved the efficiency of MNS. This agrees with the results of this experiment, where N retention has been improved using multipurpose trees as a supplement in ruminant rations [31,41,57]. 

The linear increase in the VFA concentrations in the rumen liquor with the incorporation of foliage of *B. alicastrum* into the ration may be associated with the high degradation rate of *B. alicastrum* and the increase in the degradation rate of *M. maximus*. Concentrations of VFAs found in this study were higher than the values reported by Hindrichsen et al. [57] and Muinga et al. [41] when they used foliage of *Leucaena leucocephala* as a supplement. These differences might be related to the high rumen degradation of *B. alicastrum* compared with Leucaena and to the increase in the degradation rate of the DM, OM and NDF of *M. maximus*. The absence of changes in the rumen fermentation pattern with supplementation with *B. alicastrum* in this experiment are consistent with reports by Hindrichsen et al. [57] and Muinga et al. [41].

The linear increase in the outflow and turnover rates recorded with the incorporation of *B. alicastrum* into the ration could possibly be explained by the increase in water consumption (Table 2). This increase in water consumption was probably ws associated with changes in osmotic pressure of the rumen liquid. Volatile fatty acids [50] and N-NH_3_ [46] contribute to increased osmotic pressure in the rumen. At high rumen osmotic pressure, ruminants increase water consumption [58]. In this experiment, the VFA and N-NH_3_ concentrations in the rumen liquid were increased, as well as water consumption. High quantities of liquid in the rumen increase the dilution rate and rumen outflow in sheep [50], and these results show that the dilution rate had a positive relationship with water consumption (r^2^ = 0.77). Furthermore, the outflow rate and turnover rate were positively related to DM intake r^2^ = 0.75; r^2^ = 0.83, respectively. Bonsi et al. [40] found a significant effect on the kinetics of liquid in sheep supplemented with forage of *Sesbania sesban*, which has a similar rumen degradation to *B. alicastrum*; however, differences in rumen volume were not observed. 

The improvement in the apparent digestibility of DM, NDF and OM as foliage of *B. alicastrum* was increased in the ration may be caused by a reduction in NDF content of the whole ration, as NDF has a negative effect on apparent digestibility [35]. Effective rumen degradability of the rations could have been increased by the incorporation of *Brosimum alicastrum*, considering the proportion of each ingredient in the ration and their effective rumen degradation. Assuming that the molasses was fermented completely in the rumen and taking into consideration the values for *k*_1_, the effective rumen degradability was 36.4%, 39.21%, 44.38% and 45.75% for 0%, 15%, 30% and 45% of *B. alicastrum*. This may help to explain why digestibility tended to increase; the small difference among the rations with 30% and 45% *Brosimum alicastrum* was closer to the estimated values of effective rumen degradation. The high rumen degradation of foliage of *Brosimum alicastrum* (Table 2) could be contributing to better apparent digestibility of the ration. Bonsi et al. [40] found a greater effect in digestibility with *Sesbania sesban* than with *Leucaena leucocephala*, attributing this effect to the higher rumen degradation of sesbania. Furthermore, the increase in the rumen degradation rate of NDF, DM and OM of *M. maximus* (Table 4) could have influenced apparent digestibility. Orskov et al. [9] observed a positive relation between apparent digestibility and rumen degradation rate. 

## 5. Conclusions

Results from this experiment demonstrate that the foliage of *Brosimum alicastrum* is a good source of nutrients for Pelibuey hair sheep, increasing dry matter intake, rumen ammonia-N concentration, outflow from the rumen, efficiency of microbial nitrogen synthesis and microbial nitrogen supply to the small intestine when incorporated into low-quality basal hay rations of tropical grass and may therefore be considered as a valuable feed resource for the development of sustainable systems of sheep production in the tropical areas of Latin America.

## Figures and Tables

**Table 1 animals-14-01144-t001:** Chemical composition (% DM) of *Brosimum alicastrum* and *Megathyrsus maximus* and of the experimental rations containing increasing levels of *B. alicastrum* foliage fed to Pelibuey sheep.

Item	Forage			Level of *B. alicastrum* (% DM)			
	*Brosimum* *alicastrum*	*Megathyrsus* *maximus*	*Molasses*	0	15	30	45
DM *	89.3	92.1	78.1	90.4	90.2	90.3	89.9
Organic matter	90.4	89.4	85.2	89.0	89.2	89.3	89.4
Crude protein	15.7	5.6	4.1	5.1	6.8	8.2	9.6
Neutral detergent fiber	37.5	80.0	−	72.5	66.7	60.9	55.2
Acid detergent fiber	28.5	48.4	−	43.9	41.2	38.5	35.8
Hemicellulose	9.0	31.6	−	28.6	25.5	22.5	19.4
Lignin	12.0	17.8	−	15.4	15.3	14.5	13.8

* As percentage of wet weight; DM, Dry Matter.

**Table 2 animals-14-01144-t002:** Rumen degradation (%) of DM, OM, CP and NDF of *Brosimum alicastrum* foliage in two Zebu bulls (Values are means of feed component degradation from bags incubated in the rumen in triplicate in each bull).

Item	DM	OM	CP	NDF
*A*	21.2	20.5	33.8	ND
*B*	65.7	66.1	61.4	ND
*a*	27.3	23.4	35.5	0.23
*b*	59.5	63.2	59.7	71.9
PD	86.8	86.6	95.2	72.1
*c*, h^−1^	0.11	0.11	0.12	0.09

DM, Dry Matter; OM, Organic Matter; CP, Crude Protein; NDF, Neutral Detergent Fiber; *n* = 4; ND = Not Determined; *p* = *a* + *b* (1 − e^−ct^) [10], where PD = (*a* + *b*) is potential degradability; *a* is the intercept of the curve; *b* is the asymptote of equation; *c* is the fractional degradation rate of *b* and *t* is time of incubation; *A* is zero time washing loss; *B* is the insoluble but degradable fraction = (*a* + *b*) − (*A*).

**Table 3 animals-14-01144-t003:** Voluntary intake of rations containing increasing levels of *Brosimum alicastrum* foliage by Pelibuey sheep.

Item	Level of *B. alicastrum* (% DM)				SEM	Response ^a^
	0	15	30	45		
DM intake g/d						
*Megathyrsus*	463	653	701	654	68.1	L **
*Brosimum*	0.00	115	301	535	26.7	L **
Molasses	48.0	79.7	104	123	-	-
Total (g/day)	511	848	1106	1313	106	L **
DM g/kg^0.75^	35.1	54.7	70.2	84.4	5.80	L ***
OM g/d	464	758	1032	1191	80.0	L ***
OM g/kg^0.75^	31.9	48.8	65.7	76.5	4.1	L **
NDF g/d	364	552	675	771	55.0	L **
NDF g/kg^0.75^	25.1	35.6	42.9	49.7	2.9	L ***
Water consumption	2.58	2.71	4.98	4.96	0.65	L *

^a^ L = Linear response; * *p* < 0.05; ** *p* < 0.01; *** *p* < 0.001. DM, Dry Matter; OM, Organic Matter; NDF, Neutral Detergent Fiber; ME, Metabolic Energy; MJ, Megajoules; BW, Body Weight; SEM, Standard Error of the Mean.

**Table 4 animals-14-01144-t004:** Rumen degradation constants (%) of DM and OM of *Megathyrsus maximus* incubated in Pelibuey sheep fed increasing levels of *Brosimum alicastrum* foliage.

Item	Level of *B. alicastrum* (% DM)				SEM	Response ^a^
	0	15	30	45		
Dry matter						
A	10.9					ND
a	5.09	4.36	4.66	1.69		ND
b	37.2	36.7	36.9	38.3	0.60	NS
PD	42.3	41.1	41.6	40.0	0.60	NS
c h^−1^	0.029	0.035	0.036	0.044	0.40	NS
Effective degradability	29.8	26.9	26.7	24.6	0.70	L **
Organic matter						
a	2.31	3.1	1.52	0.76		ND
b	39.2	37.5	38.3	37.8	0.65	NS
PD	41.5	40.6	39.9	38.6	0.80	NS
c, h^−1^	0.029	0.034	0.035	0.039	0.40	NS
Effective degradability	27.9	24.8	24.2	22.5	0.80	L **
NDF						
a	4.10	3.60	3.80	2.10	NS	0.13
b	32.5	33.2	32.7	32.3	NS	1.4
PD	36.6	36.8	36.5	34.3	NS	1.3
c, h^−1^	0.032	0.033	0.032	0.048	L *	0.003
Effective degradability	23.9	20.9	20.3	17.5	L **	1

^a^ NS, Not Significant; ND, Not Determined; NDF, Neutral Detergent Fiber; DM, Dry Matter; OM, Organic Matter; SEM, Standard Error of the Mean; L, linear; * *p* < 0.05. ** *p* < 0.01. *p* = *a* + *b* (1 − e^−*ct*^) [10], where PD = (*a* + *b*) is potential degradability; *a* is the intercept of the curve; *b* is the asymptote of equation; *c* is the fractional degradation rate of *b* and *t* is time of incubation; *A* is zero time washing loss; *B* is the insoluble but degradable fraction = (*a* + *b*) − (*A*).

**Table 5 animals-14-01144-t005:** Kinetics of solid digesta and liquids through the gastrointestinal tract of Pelibuey sheep fed increasing levels of *Brosimum alicastrum* foliage.

Item	Level of *B. alicastrum* (% DM)					
	0	15	30	45	SEM	Response ^a^
Solid digesta						
*k*_1_ (per h)	0.0147	0.0275	0.028	0.0412	0.003	L ***
*k*_2_ (per h)	0.0385	0.0725	0.063	0.076	0.009	L *
TT (hours)	3.84	3.54	2.99	2.63	0.34	L *
Liquids						
Rumen volume (L)	10.9	9.08	10.5	12.6	1.08	NS
Outflow rate (L/h)	0.98	1.03	1.52	2.51	0.25	L **
Turnover rate (times/d)	2.15	2.73	3.49	4.78	0.50	L **

^a^ L, Linear response; NS, Not Significant; DM, Dry Matter; SEM, Standard Error of the Mean; * *p* < 0.05; ** *p* < 0.01; *** *p* < 0.001. *k*_1_, Rate of passage of solids through the rumen (%/h); *k*_2_, Rate of passage of solids through the caecum and proximal colon (%/h); TT, Transit time in h through the gastrointestinal tract.

**Table 6 animals-14-01144-t006:** Concentration of ammonia-N (mg/100 mL rumen liquor) and VFAs in the rumen of Pelibuey sheep fed increasing levels of *Brosimum alicastrum* foliage.

	Level of *B. alicastrum* (% DM)					
Time after Feeding (h)	0	15	30	45	SEM	Response ^a^
pH	6.69	6.55	6.52	6.61	0.07	NS
N-NH_3_ (mg/100 mL)						
0	8.6	11.7	13.5	18.5	1.9	L *
3	7.56	11.7	12.7	17.6	1	L ***
6	5.43	9.82	11.4	15.8	1.2	L **
9	4.76	8.32	11.5	16.6	2	L **
Mean	6.58	10.38	11.27	17.12		
VFAs (mmol/L)	99.7	116.6	118.9	133.7	7.3	L *
VFAs mol/100 mmol)						
Acetic acid	74.8	73.1	73.4	72.2	1.4	NS
Propionic acid	16.4	17.7	16.6	17.3	0.5	NS
Butyric acid	7	7.4	7.9	8.6	0.6	NS
Others ^b^	1.7	1.8	1.9	1.8	0.31	NS

^a^ L, Linear response; * *p* < 0.05; ** *p* < 0.01; *** *p* < 0.001; ^b^ sum of isobutyrate, valerate, isovalerate. VFA, Volatile Fat Acids; N-NH^3^, Ammoniacal Nitrogen; NS, Not Significant; DM, Dry Matter; SEM, Standard Error of the Mean.

**Table 7 animals-14-01144-t007:** Urinary excretion of purine derivatives and microbial N supply to the small intestine and nitrogen balance in Pelibuey sheep fed increasing levels of *Brosimum alicastrum* foliage.

Item	Level of *B. alicastrum* (% DM)				SEM	Response ^a^
	0	15	30	45		
Allantoin (mmol/d)	2.29	4.25	6.42	7.24	0.76	L **
Uric acid (mmol/d)	1.14	1.72	2.66	3.65	0.26	L ***
Xanthine + Hypoxanthine (mmol/d)	0.2	0.26	0.41	0.45	0.07	L *
Total (mmol/d)	3.63	6.23	9.5	11.34	0.91	L ***
MNSSI (g/d)	2.2	4.92	7.93	9.72	0.89	L ***
N intake (g/d)	4.7	10.1	16	21.1	0.9	L **
N excretion (g/d)						
Fecal	3.3	5.1	6.5	9.2	0.6	L **
Urinary	3.1	4.3	5.9	7.3	0.55	L **
Total (g/d)	6.4	9.4	12.4	16.5	0.57	L **
N retention (g/d)	-1.7	0.6	3.5	4.6	0.7	L ***

^a^ L, Linear response; SEM, Standard Error of the Mean; * *p* < 0.05; ** *p* < 0.01; *** *p* < 0.001; MNSSI, Microbial N supply to the small intestine.

**Table 8 animals-14-01144-t008:** Apparent digestibility (%) of DM, OM and NDF in Pelibuey sheep fed increasing levels of foliage of *Brosimum alicastrum* tree.

	Level of *B. alicastrum* (% DM)					
Item	0	15	30	45	SEM	Response ^a^
DM	35.7	45.1	46.9	49.6	1.7	L **
OM	40.5	49.2	49.4	52	3.5	NS
NDF	33.2	44.4	45.9	47.9	3.1	L *

^a^ L, linear response; NS, Not Significant; SEM, Standard Error of the Mean; DM, Dry Matter; OM, Organic Matter; NDF, Neutral Detergent Fiber; * *p* < 0.05; ** *p* < 0.01.

## Data Availability

The data presented in this study are available on request from the corresponding author. The data are not publicly available due to the institutional policies of the University of Yucatan.

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
