# Peer review of "Effect of Brosimum alicastrum Foliage on Intake, Kinetics of Fermentation and Passage and Microbial N Supply in Sheep Fed Megathyrsus maximus Hay"

_animals, 2024, doi:10.3390/ani14081144_

Round 1
Reviewer 1 Report
Comments and Suggestions for Authors
To improve the quality of publication, I suggest the following:
1. Line 61 - you must indicate the age of the animals.
2. Line 66 - you must indicate the age of the bulls.
3.Tables 5 and 6 indicate sample size.
4. References: № 1, 31 and 32, 40 do not indicate the year of publication.
5. Lines 164-166 – the method for determining rumen pH and rumen concentration of ammonia (NH3-N) must be moved to section 2.6. Rumen pH, volatile fatty acid and ammonia concentrations (lines 129-136).
6. Line 415 – reference is made to Peter et.al (45). Under this number in the list of references there is another author.
7. Out of 57 sources, only 3 sources are from the last 5 years.
I believe that this work can be published after corrections to minor methodological errors and adding relevant links for the last 5-7 years.
Author Response
Revisor 1
- Línea 61 - debes indicar la edad de los animales. HECHO
- Línea 66 - debes indicar la edad de los toros. HECHO
3. Las tablas 5 y 6 indican el tamaño de la muestra. HECHO
Comentario del autor : Para la Tabla 5, se tomaron 4 muestras de licor ruminal (20 ml) por día para análisis de pH, AGV y amoníaco de cada animal por cada período, lo que da 16 muestras por día, multiplicado por cuatro períodos = 64 muestras para cada variable. Para la Tabla 6, se tomaron 9 muestras de licor ruminal (4 ml) para flujo líquido por animal, lo que da 36 muestras, multiplicado por cuatro períodos = 144 muestras para flujo líquido, mientras que para digesta sólida se tomaron 12 muestras de excreta fresca (40 g). directamente del recto de cada animal, lo que da 48 muestras por período y 192 muestras para todo el experimento.
- Referencias: № 1, 31 y 32, 40 no indican el año de publicación. HECHO
- Líneas 164-166: el método para determinar el pH ruminal y la concentración de amoníaco (NH3-N) ruminal deben trasladarse a la sección 2.6. pH ruminal, concentraciones de ácidos grasos volátiles y amoníaco (líneas 129-136). HECHO
- Línea 415 – se hace referencia a Peter et.al (45). Bajo este número en la lista de referencias aparece otro autor. HECHO
- De 57 fuentes, sólo 3 son de los últimos 5 años. HECHO
Comentario del autor : el experimento se realizó hace 20 años, cuando el tema de estudio (efecto de los forrajes sobre la degradación ruminal y la cinética de paso) se seguía en varios laboratorios de Asia y África, hoy en día hay menos fuentes sobre este tema. .
Creo que este trabajo se puede publicar después de corregir errores metodológicos menores y agregar enlaces relevantes durante los últimos 5 a 7 años.

Reviewer 2 Report
Comments and Suggestions for Authors
1. Line 71-72 Brosimum alicastrum foliage was harvested from mature trees, harvested foliage was sun-dried on a concrete floor for two days. Correct the grammatical issues in this sentence.
2. Line 96 The pore diameter of 53 μm is already quite large. What is the basis for selection?
3. Line 138 “2.7. Rate of passage of liquids and solids. Forty grams of Cr-mordanted fiber were introduced through the rumen cannula and fecal samples were taken at 0, 6, 8, 10, 12, 16, 24, 36, 48.72, 96 and 120 h after dosing to describe the excretion curve of the marker”. Installing rumen and duodenal fistulas is the most effective method for assessing the outflow velocity of rumen fluid and solid digesta. In the experiment, the author did not install duodenal fistulas in animals but instead relied on the excretion of Cr in feces and mathematical models to estimate the outflow velocity. The main challenge during the collection of feces over the 0-120 hour period is ensuring that fresh samples are obtained at each time point. This issue would be resolved if animals were equipped with duodenal fistulas. The calculation of Microbial N supply also has this problem.
4. Line 158-159 “NDF and ADF were determined as described by Goering and Van Soest”. Is amylase added during the determination of NDF? Does NDF correct the ash content?
5. Line 165-166 “Concentration of ammonia (NH3-N) in rumen liquor was determined with a specific ion electrode (Corning Ammonia Electrode Cat. No.476130), VFA were analyzed by gas chromatography”. Add additional references for the ammonia nitrogen determination method. Include details about the specific models of equipment and chromatographic columns employed in gas chromatography. Could you confirm whether an internal standard was utilized in the determination of VFA?
6. Line 172-176 Supplementary statistical models.
7. Table 2 what is SE?
8. Line 228 SEM's annotation is wrong
9. Line 93 How many animals were used to determine the ruminal degradation kinetic parameters of the two feeds. Why do the ruminal degradation kinetic parameters of B. alicastrum have no statistical results, while M.maximus does? This part is a bit confusing.
Author Response
Reviewer 2
- Line 71-72 Brosimum alicastrum foliage was harvested from mature trees, harvested foliage was sun-dried on a concrete floor for two days. Correct the grammatical issues in this sentence. DONE
- Line 96 The pore diameter of 53 μm is already quite large. What is the basis for selection? DONE
Author’s comment: this (53 µm) is the standard pore size for bags in rumen incubation studies in many laboratories around the world. In addition, we have tested several pore size for rumen incubation bags in our laboratory and that pore size gave the lower residual standard error for the kinetic parameters, suggesting a better statistical fit for the data. In fact, that pore size, is the size of the bags supplied by Bar Diamond Inc. (Parma, Idaho, USA) a major commercial supplier of bags in the world.
- Line 138 “2.7. Rate of passage of liquids and solids. Forty grams of Cr-mordanted fiber were introduced through the rumen cannula and fecal samples were taken at 0, 6, 8, 10, 12, 16, 24, 36, 48.72, 96 and 120 h after dosing to describe the excretion curve of the marker”. Installing rumen and duodenal fistulas is the most effective method for assessing the outflow velocity of rumen fluid and solid digesta. In the experiment, the author did not install duodenal fistulas in animals but instead relied on the excretion of Cr in feces and mathematical models to estimate the outflow velocity. The main challenge during the collection of feces over the 0-120 hour period is ensuring that fresh samples are obtained at each time point. This issue would be resolved if animals were equipped with duodenal fistulas. The calculation of Microbial N supply also has this problem. DONE
Author’s comment: We disagree on the point of a most effective method for assessing outflow rate from the rumen (by using duodenal cannulas). Most techniques have advantages and disadvantages. In this work we used a widely accepted technique to measure outflow rate (see Faichney, 1984) which is the use of Cr-mordanted fibre (Uden et al. 1980) and the mathematical model of Grovum and Williams (1973). These techniques are widely employed by animal scientists throughout the entire world to measure such kinetic variables because they comply with the basic principles for selecting a marker and offer the possibility of relatively compare the rate of flow of solid digesta through the gastrointestinal tract. In fact, the results found are logical and represent a plausible explanation for the response variables under study. We in fact collected fresh fecal matter by sampling directly from the rectum at the given time points (this has been corrected in the text). We thank the reviewer for the point. The estimation of microbial N supply to the small intestine with the purine derivative technique is a method widely employed to measure such a variable. It compares pretty well with the purines in duodenal digesta technique. In fact, a total collection of urine (24 h) is carried out in metabolic crates and the urine is taken directly from the urine container. No under or overestimation of microbial nitrogen supply has been reported with this technique so far, therefore is quite appropriate because it does not require the use of duodenally cannulated animals for sampling.
- Line 158-159 “NDF and ADF were determined as described by Goering and Van Soest”. Is amylase added during the determination of NDF? Does NDF correct the ash content? DONE
Author’s comment: We did not use α-amylase because the rations were composed basically of forages (Brosimum alicastrum and Megathyrsus maximus) containing no starch at all in the samples. In our laboratory we do not use this enzyme anyway, but rather we use pancreatin and we incubate the samples containing starch during 4 h at 37 ⁰C. For NDF analysis we use the Ankom analyzer which does not correct for ash content of the sample, but rather determine ash by incinerating the sample in a muffle furnace at 600 °C. In other words, we do another chemical determination for ash content in the sample.
- Line 165-166 “Concentration of ammonia (NH3-N) in rumen liquor was determined with a specific ion electrode (Corning Ammonia Electrode Cat. No.476130), VFA were analyzed by gas chromatography”. Add additional references for the ammonia nitrogen determination method. Include details about the specific models of equipment and chromatographic columns employed in gas chromatography. Could you confirm whether an internal standard was utilized in the determination of VFA? DONE
Author’s comment: We added a new reference for the ammonia nitrogen determination method used by Galyean and Chabot (1981; Journal of Animal Science; 52:1197-1204) who employed a specific ion ammonia electrode instrument to determine ammonia concentration in the rumen of cattle. We added the gas chromatograph (Hewlet Packard) and the column fitted to the apparatus. No internal standard was used in the determinations.
- Line 172-176 Supplementary statistical models. DONE. The statistical model was added.
- Table 2 what is SE? DONE
Author’s comment: SE stands for standard error of the rate of degradation (c) only. We concluded that it was better to delete it altogether to avoid confusion by the reader, Data are the means of the degradation of feed components incubated by triplicate in the rumen of two Zebu bulls. There was no point in incorporating the SE of every kinetic parameter derived, which was low with the use of the standard incubation procedure (all bags inserted at the beginning and withdrawn from the rumen according to chosen incubation hours).
- Line 228 SEM's annotation is wrong DONE
- Line 93 How many animals were used to determine the ruminal degradation kinetic parameters of the two feeds. Why do the ruminal degradation kinetic parameters of B. alicastrum have no statistical results, while M. maximus does? This part is a bit confusing. DONE
Author’s comment: To estimate the rumen degradation parameters of Brosimum alicastrum, we used two Zebu (Bos indicus) bulls cannulated in the rumen. We incubated the nylon bags on those two animals because of the larger rumen capacity to hold the bags and the uncomplicated insertion and withdrawal of bags with large cannulas (10 cm diameter). No statistical treatment of the data was envisioned because the purpose was only to describe the kinetics of rumen degradation of chemical components (DM, CP, OM, etc.) and no comparisons were planned. On the other hand, kinetics of rumen degradation of Megathyrsus maximus grass was carried out in four rumen cannulated Pelibuey hair sheep. A Latin square design was used, comparisons among treatments were planned and the data was analyzed by ANOVA with SAS.

Reviewer 3 Report
Comments and Suggestions for Authors
See attachment

Author Response
Reviewer 3
The present manuscript presents result from feeding trials with cannulated ruminants fed by diets with increasing quantities of dried leaves from a common tree. The hypothesis is that the tree leave hay is highly digestible and promotes the microbial protein synthesis in the rumen. Hay from tree leaves disappeared from feeding tables but in several parts of the world this type of biomass provides the potential to enrich the resources for animal production.
- The experimetal design is valid.
- The trials include a sufficient number of animals
- The analytical procedures are correct.
- Sequence of topics: In the abstract and further sections the authors start by detail (e.g. row 22: rate of degradation/fermentation) and consider later feed intake etc. This makes the manuscript difficult to read. My recomendation is to reorganize the structure in presenting the results and the discussion of the results, e.g. a) feed/nutrient and water intake, b) ruminal digestion, c) microbial protein etc. DONE. Thanks for the suggestion.
- Abstract: Please add a comment about the beneficial rate of inclusion for Bros. alic. DONE
- Please give a comment why using Zebu bulls instead of sheep for measuring rumen fermentation DONE
Author’s comment: We choose the use of cattle to measure rumen kinetic parameters of Brosimum alicastrum due to their larger capacity to hold a large number of incubation bags in the rumen compared to sheep. We incubated the bags by triplicate at each time point, so a large number of bags was incubated in the rumen, the larger capacity of the rumen of cattle makes the work of inserting and withdrawing bags easier to the students doing the sampling. From previous experiments in this laboratory, no large differences are expected in the estimation of rumen kinetic parameters by using cattle or sheep, as long as similar basal rations of tropical grass are fed to the experimental animals.
- Row 132 ff: ref. PEG and Cr.mordanted fiber: pleas give the information regarding the application of markers related to feeding DONE
Author’s comment: Those markers, PEG (for liquid phase) and Cr-mordanted fibre (for solid phase) have been widely employed in animal science research for more than 40 years (at least). The procedures have been described in scientific papers to some detail and in the proceedings of the International Symposium on Ruminant Physiology for years, and they are modified and improved, or new markers appear. The methods of estimation are also known and most of them use regression techniques in the attempt to identify the pools (rumen, small intestine, large intestine, etc.) and there are complex models with several compartments (up to five). The Grovum and Williams (1973) model (used in this paper), is one of the simplest with only two compartments (rumen and large intestine), therefore it is easier when trying to interpret the data and the differences among treatments.
- Row 202; please improve the title of the table. The table includes the parameter for the model but not simply that what is considered in the title DONE
- Tab. 3: what ist the reason for giving data in g/day and per kg BM^0.75 and finally ME per kg BM. This is confusing. Please make this table more compact and please exchange water intake by water consumption because the latter is really measured and consider to add the ratio of water per g DM or even better per kg DM (DONE)
Author’s comment: Data in shown in several expressions and no attempt was made to confound the reader. On the other hand, intake is described in g/d, so the reader can quickly assess the amount of feed consumed by the animals, but also in g/kg0.75 because with this expression, the reader can also quickly assess the level of intake relative to maintenance, since it is known that hair sheep will be consuming at around 50 g DM/kg0.75 at maintenance when fed tropical grasses. Thus, one quickly can pinpoint the DM intake of the animals relative to energy balance (positive/negative). We deleted from Table 3, the line describing ME intake per kg BW, we thank the referee for this suggestion.
- Tab. 2; are these data obtained by Zebu bulls? Please make this clear. DONE, we thank the referee for this question which was truly necessary to incorporate in Table 2.
- Please uniform in title of tables: sheep or Pelibuey sheep DONE
- Again I mention the sequence of data e.g. chemical composition, ruminal digestion, feed intake; this is difficult to read. Please consider a other structure in the manuscript DONE
- I miss a discussion regarding the low protein intake/concentration of the diets. At least the 0- Bros.alic.-diet is marginal or even deficient in N and protein respectively. The energy may be ensure a higher microbial protein synthesis as calculated by urinary allantoin DONE
Author’s comment: We agree with the referee, the control ration (i.e. 0% Brosimum alicastrum) was clearly deficient in rumen degradable protein, as evidenced by the small supply of microbial nitrogen to the small intestine and the low dry matter intake. We have added a paragraph in the text to correct this. We appreciate the comment by the referee.
- The authors cite several papers that substantiate the value of compounds harvested from trees. I am convinced it would be a benefit to have table with key data for those compounds of different origin in comparison to the Bros.alic. In this study. DONE (considered with thanks)
Author’s comment: We appreciate the comment by the referee, but we think a larger discussion, let alone a table on phenolic compounds, is limited by the small number of samples analyzed and the use of foliage of only one tree in the present study. No doubt a comparable table describing the concentrations of several secondary metabolites and the possible effects on intake and fermentation would be worthwhile of a specific review on the subject matter. Thus, we believe that a table on this (phenolic compounds) would be inappropriate for this paper under the present circumstances.

Round 2
Reviewer 3 Report
Comments and Suggestions for Authors
Thanks for considering the remarks
ref. marker: I know the potential of substances used as marker and the bulk of literature on the suitability and limitations etc. My remark addresses the timing of marker application in relation to feed provision.